# Contrasting Responses of Multispatial Soil Fungal Communities of *Thuja sutchuenensis* Franch., an Extremely Endangered Conifer in Southwestern China

You-wei Zuo,[a,b] Ping He,[c] Jia-hui Zhang,[a,b] Wen-qiao Li,[a,b] Deng-hao Ning,[a,b] Yu-lian Zeng,[a,b] Ying Yang,[a,b] Chang-ying Xia,[a,b] Huan Zhang,[a,b] 🆔 Hong-ping Deng[a,b]

aCenter for Biodiversity Conservation and Utilization, School of Life Sciences, Southwest University, Beibei, Chongqing, China

bChongqing Key Laboratory of Plant Resource Conservation and Germplasm Innovation, Institute of Resources Botany, School of Life Sciences, Southwest University, Beibei, Chongqing, China

cChongqing Academy of Science and Technology, Low Carbon and Ecological Environment Protection Research Center, Liangjiang New Area, Chongqing, China

**ABSTRACT** *Thuja sutchuenensis* Franch. is an endangered species in southwest China, distributed sporadically in mountainous areas. Soil property and soil fungal community play a crucial role in plant growth and survival. Nevertheless, understanding soil properties and the soil fungal community in the areas where *T. sutchuenensis* is distributed is extremely limited. Hence, this study collected a total of 180 soil samples from five altitudinal distribution areas (altitudinal gradients) and three vertical depths throughout four horizontal distances from the base of each tree. The results found that altitudinal gradients and vertical depths altered soil properties, including pH, organic matter content, water content, total nitrogen, phosphorus, and potassium, and available nitrogen, phosphorus, and potassium. The fungal alpha diversity indexes (Chao1 and Shannon) and beta diversity were dramatically decreased with elevation. In addition, high altitudes (2,119 m) harbored the highest relative abundance of ectomycorrhizal fungi (27.57%) and the lowest relative abundance of plant-pathogenic fungi (1.81%). Meanwhile, we identified a series of fungal communities, such as *Tomentella*, *Piloderma*, *Cortinarius*, *Sebacina*, and Boletaceae, that play an essential role in the survival of *T. sutchuenensis*. The correlation analysis and random forest model identified that water content and total phosphorus showed strong relationships with fungal characteristics and were the primary variables for Zygomycota and Rozellomycota. Collectively, the findings of this integrated analysis provide profound insights into understanding the contrasting responses of *T. sutchuenensis* soil fungal communities and provide a theoretical basis for *T. sutchuenensis* habitat restoration and species conservation from multispatial perspectives.

**IMPORTANCE** The present study highlights the importance of fungal communities in an endangered plant, *T. sutchuenensis*. Comparative analysis of soil samples in nearly all extant *T. sutchuenensis* populations identified that soil properties, especially soil nutrients, might play critical roles in the survival of *T. sutchuenensis*. Our findings prove that a series of fungal communities (e.g., *Tomentella*, *Piloderma*, and *Cortinarius*) could be key indicators for *T. sutchuenensis* survival. In addition, this is the first time that large-scale soil property and fungal community investigations have been carried out in southwest China, offering important values for exploring the distribution pattern of regional soil microorganisms. Collectively, our findings display a holistic picture of soil microbiome and environmental factors associated with *T. sutchuenensis*.

**KEYWORDS** *Thuja sutchuenensis*, spatial variation, fungal community, soil properties

Address correspondence to Hong-ping Deng, denghp@swu.edu.cn.

The authors declare no conflict of interest.

Wild plants are a critical part of the ecosystem as well as a crucial strategic resource for current society (1). The extinction of a variant reduces the genetic

resources of the species and triggers a chain reaction in its survival network, leading to a series of species extinctions and even ecosystem instability (2). The genus *Thuja* includes monoecious evergreen coniferous trees belonging to the Cupressaceae and is an essential medical plant harboring antimicrobial, antibiofilm, and anticancer activities (3, 4). The bark and root of some *Thuja* plants are a traditional medicinal material, which can relieve tension and anxiety and treat coughs (5, 6). However, in the list of the top 20 endangered gymnosperm species, Cupressaceae alone account for five, with the dominant threats linked to human activities, particularly habitat transformation and harvesting (7). *Thuja sutchuenensis* Franch. is an evergreen coniferous tree and was first identified in 1892 in a limestone mountain area at an altitude of 1,400 m in Chengkou County, Chongqing Municipality, China (8). Nevertheless, the tree was "extinct" for more than 100 years and was ultimately rediscovered in October 1999. Currently, *T. sutchuenensis* has become a critically endangered species unique to southwest China (i.e., Chengkou County and Kaizhou District of Chongqing Municipality and Xuanhan County of Sichuan Province) with sporadic distributions (altitudinal range 800 to 2,100 m) (8). Hence, as a rediscovered conifer (9), it is necessary to preserve the endangered plant and promote its population prosperity comprehensively.

Among various environmental factors, the biotic and abiotic soil components are the most crucial driving factors for plant growth and survival (10). Regarding the abiotic components, the soil pH, organic matter (OM), water content (WC), and nitrogen (N), phosphorus (P), and potassium (K) contents are the primary soil property variables (11, 12). Among these, soil OM, for example, is the major nutrient input for plants and plays an essential role in enhancing soil structure, temperature stability, and ventilation (13, 14). The rhizosphere of plants is where the root system's life activities and metabolism have the most direct and decisive influence on the soil (15). In fact, it harbors a large number of microorganisms (e.g., soil fungi), which in turn promote the synthesis of humus and increase the content of soil OM (16). *Thuja* is naturally distributed in the mountains of southwest China, where the habitat is harsh, subjected to desiccation, drought, and low temperatures (17). Besides strong climatic gradients, the biotic and abiotic soil components at different spaces, both broadly (e.g., altitude) and locally (e.g., horizontal and vertical transects), are highly distinctive, further increasing the complexity of the environment. Therefore, investigation of soil properties provides a profound perspective on the living conditions and species protection of *T. sutchuenensis*.

Soil microorganisms (e.g., rhizospheric and nonrhizospheric microorganisms) include beneficial microorganisms, which induce the growth of plants, and harmful microorganisms, which vitiate the survival of plants (18). The beneficial microorganisms existing in the soil around plant roots can facilitate plant prosperity in at least two different ways. On the one hand, microorganisms can directly stimulate a systemic resistance response in plants via the regulation of jasmonic acid, ethylene, and salicylic acid pathways (19, 20). In addition, volatile organic compounds produced by rhizosphere microorganisms could mediate intricate communications between microorganisms and plants (21, 22). On the other hand, microorganisms can promote the uptake of environmental nutrients such as N, P, and K to affect plant growth. Well-known instances are the mycorrhizal fungi, which boost phosphorus uptake and nitrogen-fixing rhizobia (23, 24). Studies have shown that purposeful conservation of fungal communities in sites where endangered plants live is an efficient conservation strategy. For example, the protection of *Rhizopogon yakushimensis* synergistically protected *Pinus amamiana*, an endangered species endemic to Japan (25). Some fungi (e.g., *Trichoderma*, *Mortierella*, and *Hypocrea*) were also assumed to protect the natural habitat of the first-class endangered plant *Cypripedium japonicum* and promote its reproduction (26). Therefore, the investigation of soil fungi is of great significance for understanding the growth and survival, resistance response, and protection of rare and endangered plants.

Many environmental factors in mountain ecosystems vary dramatically along the spatial distance, including above- and below-ground microorganisms (27). Currently, growing evidence has explored the response of the microbial community in bulk soil to altitudinal gradients, indicating that altitudinal gradients distinctively alter the microbial composition and diversity of soils by affecting plant and soil properties (28, 29).

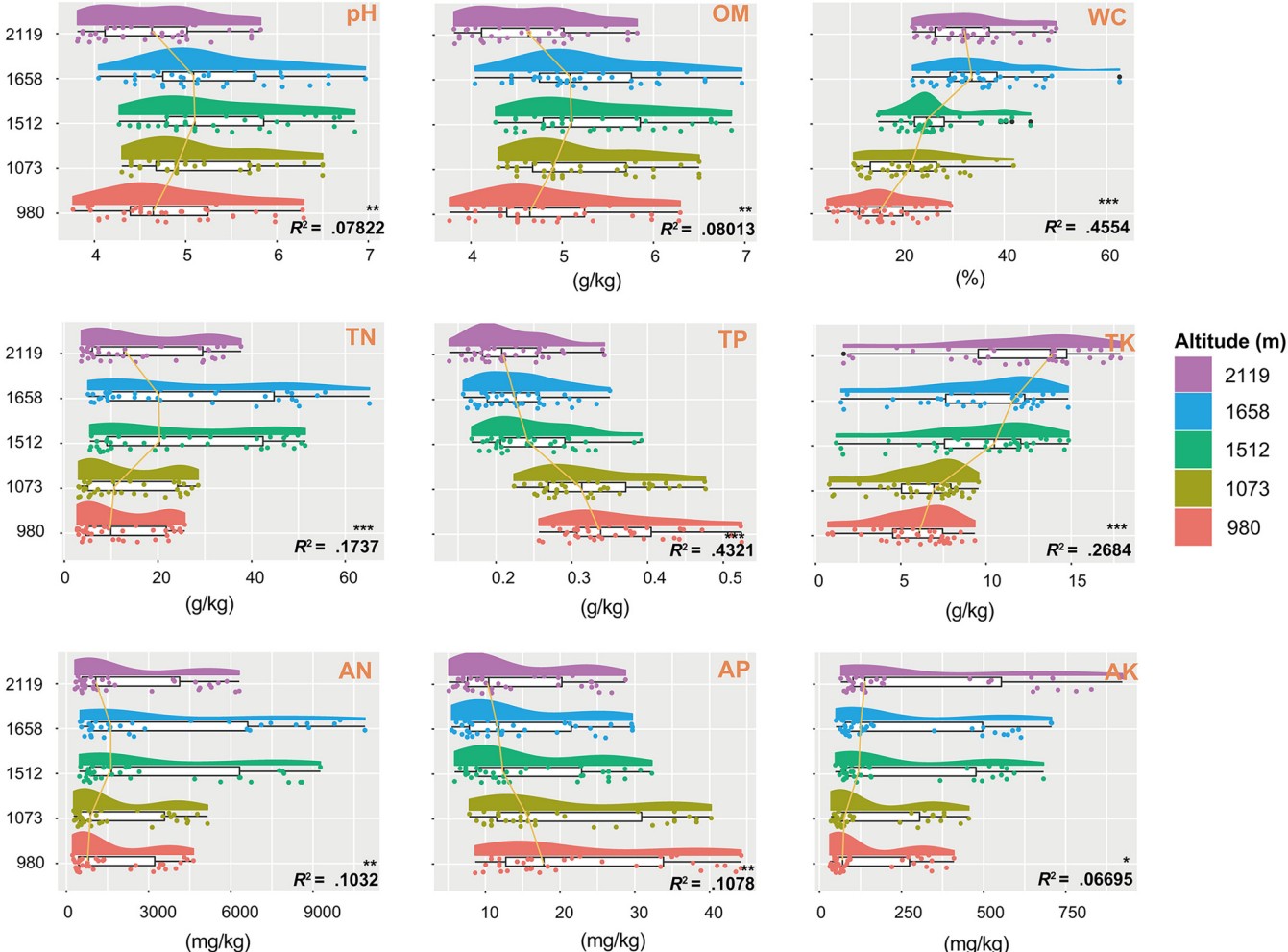

**FIG 1** Raincloud plot showing soil properties at different elevations. The half-violin diagram (cloud) shows the kernel density of the data distribution, and the scatter diagram (rain) shows the degree of dispersion. The raincloud plot also includes a box plot (umbrella) and lines (thunder) that link the medians of different groups. Abbreviations: OM, organic matter; WC, water content; TN, total nitrogen; TP, total phosphorus; TK, total potassium; AN, available nitrogen; AP, available phosphorus; AK, available potassium. Significant difference: *, $P < 0.05$; **, $P < 0.01$; ***, $P < 0.001$.

In addition, microbial communities were detected between the topsoil and deep soil, for which microbial diversity varies with vertical depth (30–32). Surrounding a tree, soil microbial communities might be affected by an environmental heterogeneity gradient along the radial distance of the tree's rhizospheric system (33). Yet, investigation of how soil microbiomes are distributed along the altitudinal gradients or on a fine scale concerning *T. sutchuenensis* root-related gradient is minimal.

To comprehensively understand the alterations in *T. sutchuenensis* root-related soil fungal community composition and species diversity along spatial variations (i.e., altitudinal gradients, horizontal distances, and vertical depths) and reveal their biotic and abiotic determinants, we collected soil samples from five altitudinal locations of *T. sutchuenensis* (altitude gradient), in each one of which three plant individuals were selected (biological replicates). Soil samples were collected at three vertical depths throughout four horizontal distances from the base of each tree, obtaining a total of 180 soil samples. Here, we hypothesized that fungal species diversity and community composition would differ according to the spatial variations and that different fungal communities would respond distinctively to spatial variations and soil properties. Specifically, we aimed to (i) reveal how soil properties respond to spatial variations, (ii) compare how soil fungal community diversity and composition respond to spatial variations, and (iii) estimate the impacts of soil properties on the soil fungal community. Collectively, this comprehensive study

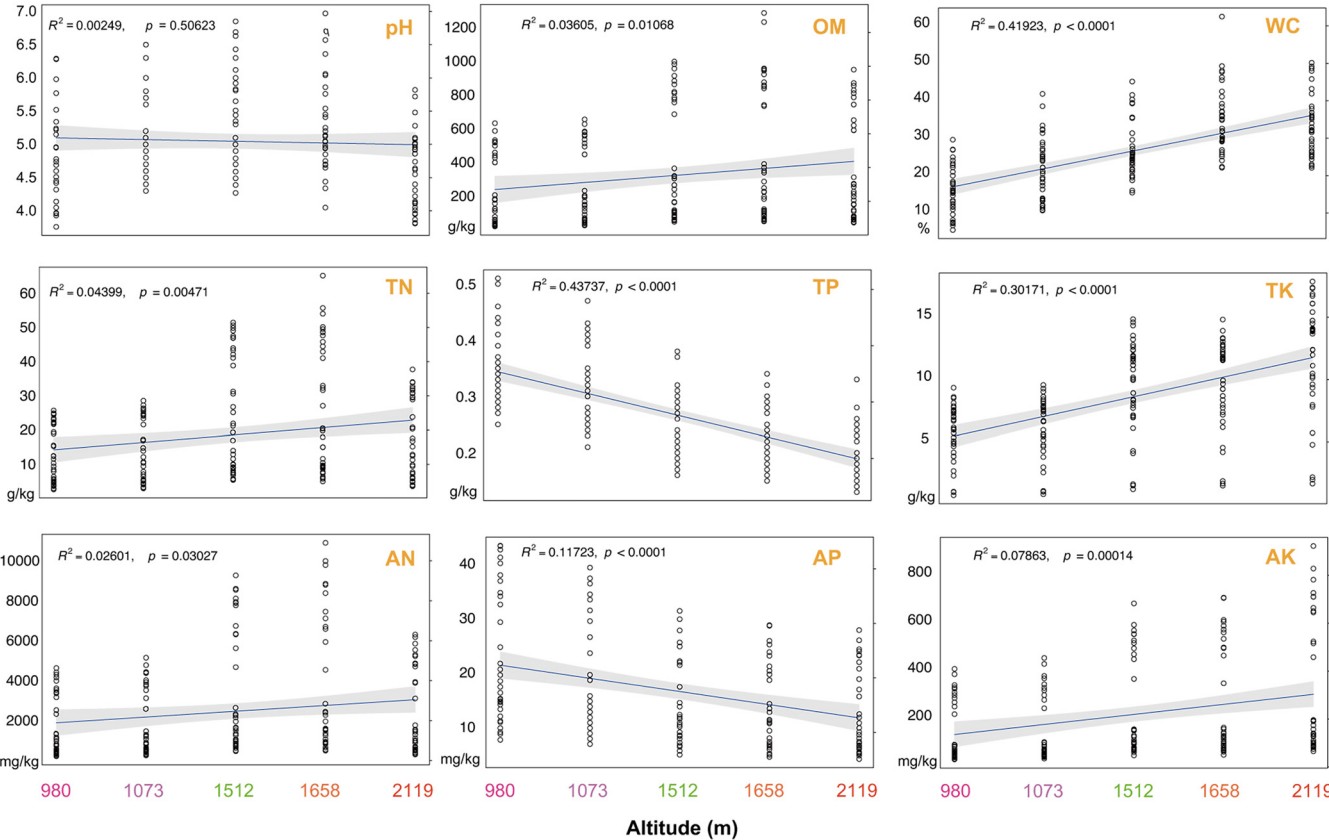

**FIG 2** Variation in soil properties with altitudinal gradient. Linear least-squares regression relationships between altitudes and soil properties were estimated. The adjusted $R^2$ was employed to determine the models that fitted with the whole altitudes.

provided reference data for understanding soil properties and fungal communities in *T. sutchuenensis* in multispatial dimensions and also generated a theoretical basis for habitat restoration and species conservation from large-scale perspectives.

## RESULTS

**Altitudinal, horizontal, and vertical variation in soil properties.** We analyzed individual soil physical and chemical properties, including pH, OM, WC, total nitrogen, phosphorus, and potassium (TN, TP, and TK, respectively), and available nitrogen, phosphorus, and potassium (AN, AP, and AK). All soil physical and chemical properties were altered significantly at different altitudes ($P < 0.05$) (Fig. 1). Specifically, soil pH ($R^2 \sim 0.08$, $P < 0.01$), OM ($R^2 \sim 0.08$, $P < 0.01$), TN ($R^2 \sim 0.17$, $P < 0.001$), and AN ($R^2 \sim 0.10$, $P < 0.01$) were the lowest at 2,119 m and 980 m and the highest at 1,658 m, 1,512 m, and 1,073 m. In addition, TP ($R^2 \sim 0.43$, $P < 0.001$) and AP ($R^2 \sim 0.11$, $P < 0.01$) had the same variation trend, and their statistical values decreased with the increase of altitude. On the contrary, WC ($R^2 \sim 0.46$, $P < 0.001$), TK ($R^2 \sim 0.27$, $P < 0.001$), and AK ($R^2 \sim 0.07$, $P < 0.05$) in soil increased with heights. When considering the four horizontal distances, none of the soil parameters, except pH and TK, which showed the highest values in sites A and B, differed (see Fig. S1 in the supplemental material). Distinctively, the nine soil indicators varied remarkably with increasing soil vertical depth (Fig. S2). In fact, the contents of OM, WC, TN, TP, AN, AP, and AK displayed a downward tendency with the depth increase, but TK showed an upward trend. In addition, we also used least-squares linear regressions to verify the soil properties at different altitudes. We demonstrated that soil indicators harbored significant variation with height among the five altitudinal plot sites (Fig. 2). The variations of OM, WC, TN, TK, AN, and AK increased with altitude, whereas TP and AP decreased with altitude.

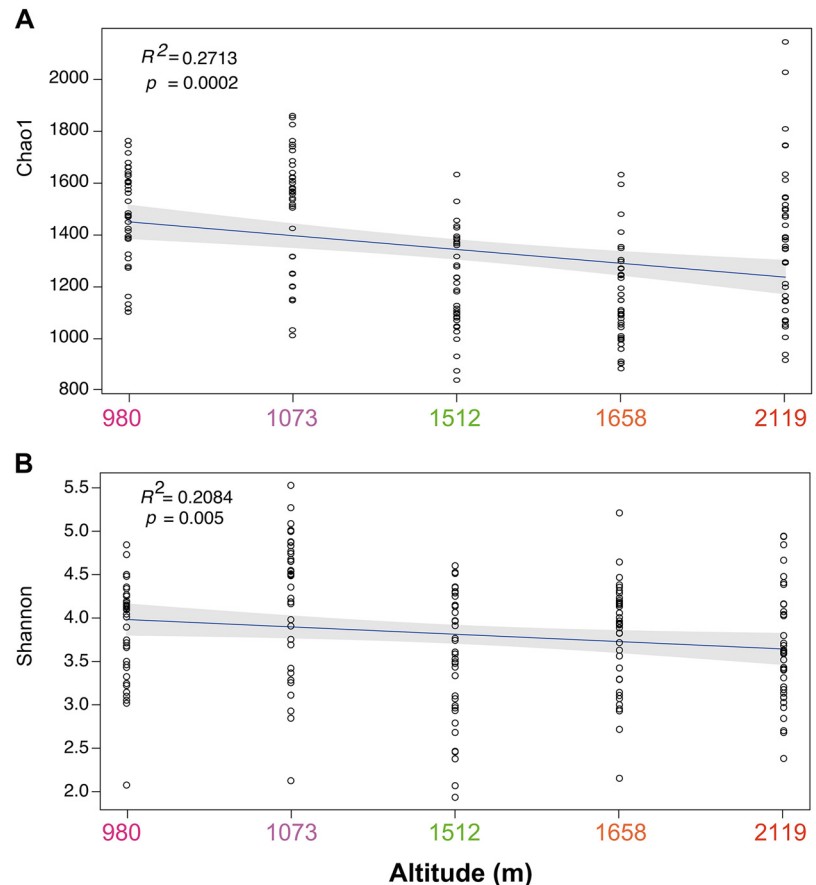

**FIG 3** General patterns of fungal alpha diversity with altitudinal gradients according to (A) Chao1 and (B) Shannon indexes.

**Diversity of fungal communities under different spatial distributions.** The alpha diversity indexes Chao1 and Shannon were used to evaluate the diversity of soil fungal communities at the sites of *T. sutchuenensis*. The results indicated that altitude was the dominant driving factor for the diversity of soil fungal communities, whereas horizontal distance and vertical depth played a limited role. In fact, fungal diversity significantly decreased with increasing altitudes according to least-squares linear regression analysis (Fig. 3) based both on the Chao1 ($R^2 \sim 0.083$, $P < 0.0001$) and Shannon ($R^2 \sim 0.029$, $P < 0.05$) indexes. However, the horizon and depth did not alter community diversity, such as Chao1 and Shannon indexes, in *T. sutchuenensis* soil according to the least-squares linear regression analysis ($P > 0.05$) (Fig. S3).

We used beta diversity analysis to explore the differences of species diversity for the three spatial variations. Nonmetric multidimensional scaling (NMDS) analysis demonstrated that the soil samples from different altitudes showed distinct clusters in the ordination space (Fig. 4A). Moreover, we estimated the differences in beta diversity among different fungal altitudinal groups according to Bray-Curtis distance, which further verified that altitude could trigger significant differences at taxonomic levels (analysis of similarity [ANOSIM]) (Fig. 4D). Interestingly, NMDS analysis showed that soil samples at a horizontal distance and vertical depth had a high degree of coincidence in the cluster diagram, with no clear differential patterns (Fig. 4B and C).

**Estimating fungal community structure along altitudinal gradients.** Based on the sequences retrieved, a total of 12,519 operational taxonomic units (OTUs) were obtained following the cutoff of 97% sequence similarity, including 1,819 OTUs that occurred in five altitudes. The majority of fungal sequences belonged to the phyla Ascomycota (47.69%), Basidiomycota (39.14%), and Zygomycota (0.42%) (Fig. S4A). At

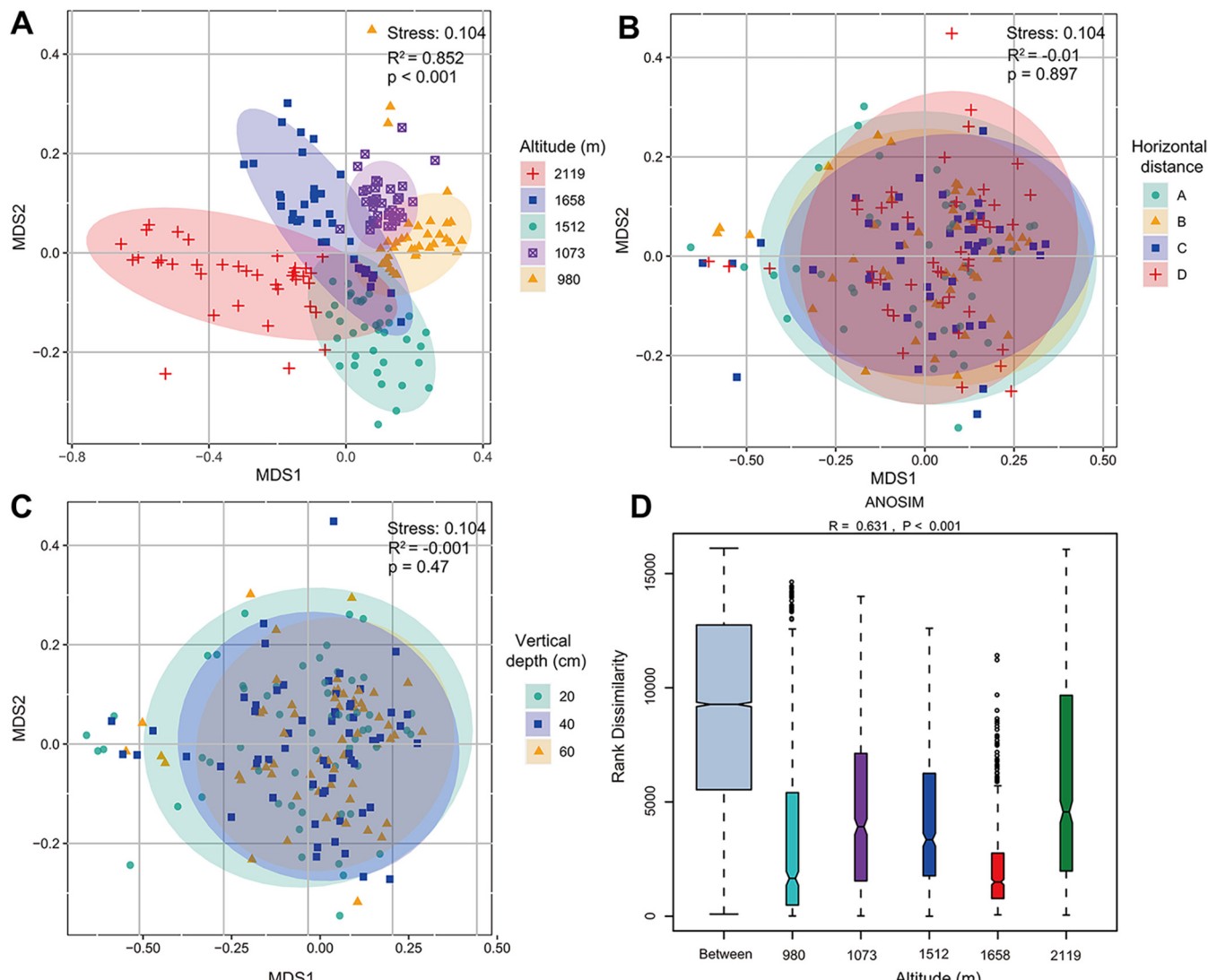

**FIG 4** Nonmetric multidimensional scaling (NMDS) analysis of fungal beta diversity from different (A) altitude gradients, (B) horizontal distances, and (C) vertical depths. Ninety-five percent confidence ellipses are displayed around the samples. (D) Alterations in beta diversity were calculated based on a Bray-Curtis distance matrix.

the genus level, the most abundant genera found were *Hygrocybe* (5.95%), *Sebacina* (3.23%), *Entoloma* (2.24%), and *Russula* (2.16%) (Fig. S4B).

We conducted functional classification for all yielded OTUs based on FUNGuild analysis and found that a total of 2652 OTUs had accurate functional groups, of which 26.96% were pathotrophs, 53.85% were saprotrophs, and 48.79% were symbiotrophs. When considering the altitudinal gradient, we found that ectomycorrhizal fungi (EMF) had the highest number of OTUs (sequence richness) and relative abundance (OTU richness) at all altitudinal gradients and peaked at 2,119 m (12.71% and 27.57%, respectively) (Fig. S5). Meanwhile, we focused on the plant-pathogenic fungi (PPF) and arbuscular mycorrhizal fungi (AMF) and found that both showed high abundances at 1,512 m (2.63% for PPF and 2.88% for AMF). Interestingly, the results showed that the abundance of PPF decreased with altitude and reached the lowest abundance at 2,119 m (1.81%).

**Contrasting responses of multispatial soil fungal genera at a fine scale.** To further identify fungal taxa responsible for community differentiation among different altitudes, horizontal distance, and soil depth of the tree, we conducted linear statistical analysis on all of the identified OTUs at the genus level and displayed the most

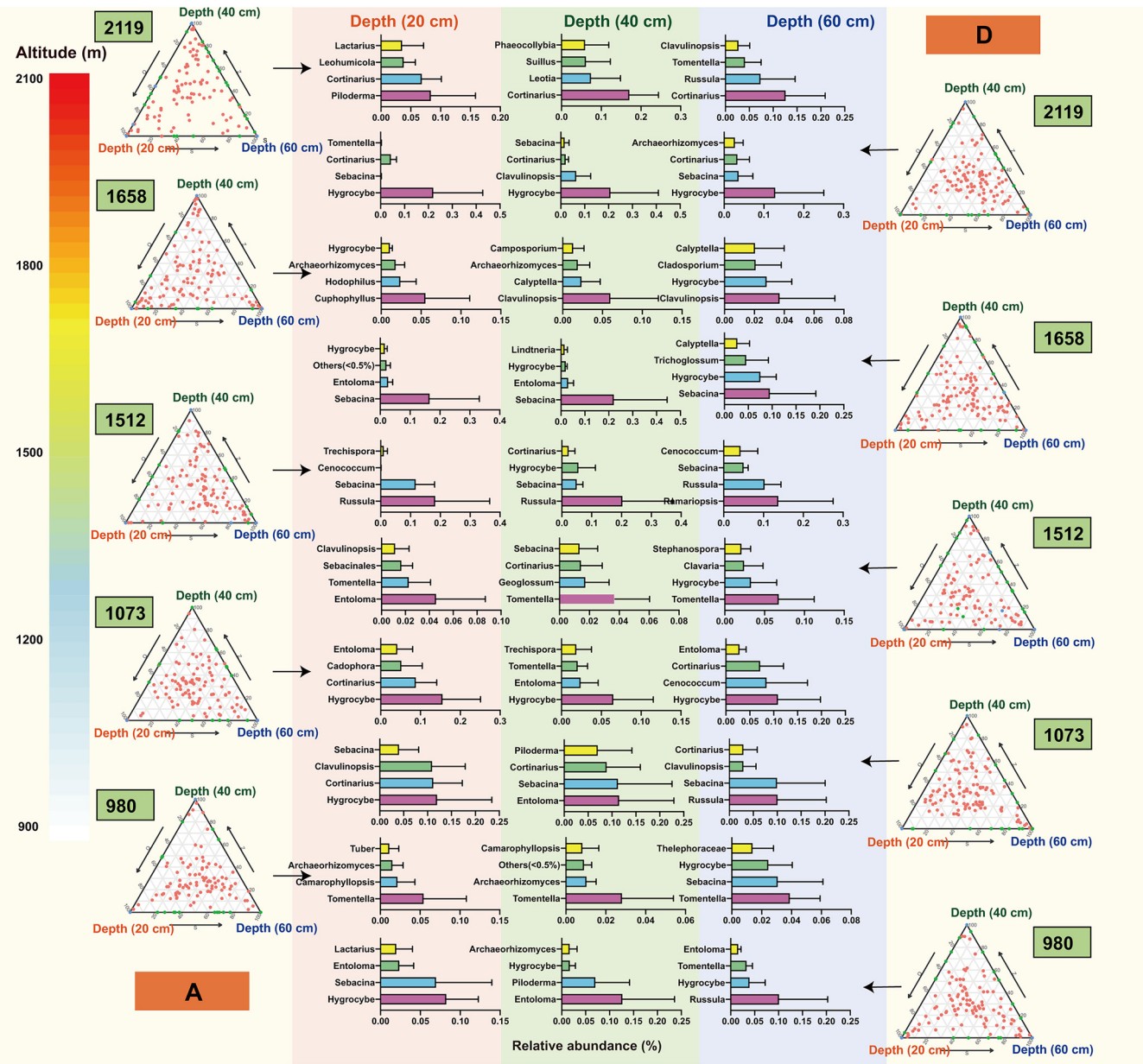

**FIG 5** Taxonomic distribution of fungal taxa (genera) accounting for community differentiation among different spatial variations. The base of the trunk is considered the rhizosphere soil (site A in the left panel) and the bare area the nonrhizosphere soil (site D in the right panel). Ternary plots display the distributions of the identified differentiation taxa. Each data point represents 1 OTU. The most abundant four genera are presented in bar plots. The colors of points indicate the OTUs are dramatically enriched among distinct depths.

abundant four genera at a fine scale (Fig. 5; Fig. S6). In this study, we identified the soil from the base of the trunk (site A) as the rhizosphere soil of *T. sutchuenensis* and the soil in bare land (site D) as the nonrhizosphere soil (Fig. 5). The linear analysis showed a strong correlation between the rhizosphere and nonrhizosphere soils, indicating that the soil was highly conservative in the horizontal distance (Fig. S6). Furthermore, vertical samples with depths of 0 to 20 cm, 20 to 40 cm, and 40 to 60 (labeled "20 cm," "40 cm," and "60 cm" on the figure) were taken for each horizontal site. We found that at low altitudes (980 m), *Tomentella* was the rhizosphere fungal genus with the highest relative abundance in all three depths. At an altitude of 1,073 m, *Hygrocybe* dominated the rhizosphere in all three depths. At 1,512 m and 1,658 m, the dominant rhizosphere fungal genera changed to *Russula* (20 cm and 40 cm) and *Clavulinopsis* (40 cm and 60 cm), respectively. At 2,119 m, *Cortinarius* was the rhizosphere fungal genus with the

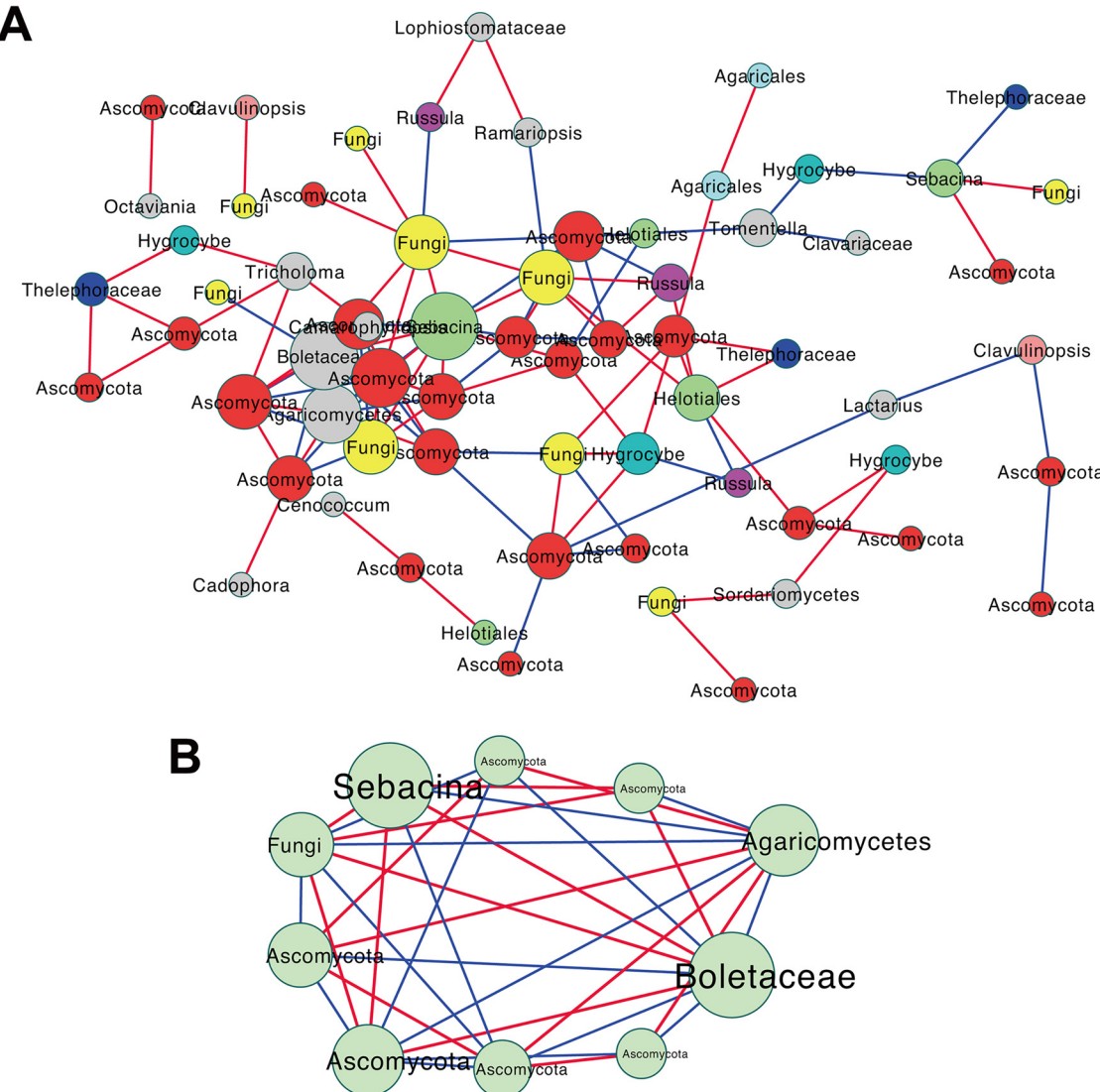

**FIG 6** Co-occurrence analysis of fungal community. (A) Associated network of co-occurrence OTUs. The red line shows a positive correlation; the blue line shows a negative correlation. The colors of the nodes indicate different phyla. Phyla with a relative abundance of less than 2% are indicated by gray nodes. The size of nodes indicates the stress of interaction. (B) Most densely closed connected module extracted from panel A based on the MCODE analysis.

highest relative abundance (40 cm and 60 cm) (Fig. 5, left panel). Collectively, the changes of dominant rhizosphere fungal genera with elevations found were as follows: *Tomentella* (980 m)→*Hygrocybe* (1,073 m)→*Russula* (1,512 m)→*Clavulinopsis* (1,658 m)→*Cortinarius* (2,119 m). For samples between rhizosphere and nonrhizosphere soils (sites B and C) (Fig. S6) and in nonrhizosphere soils (site D) (Fig. 5, right panel), the distributions of the identified indicator OTUs were intricate. None of the most dominant fungal genera showed a similar pattern in sites B, C, and D, with site A above 1,512 m. For instance, *Cortinarius* (site A), *Elaphomyces* (site B), *Russula* (site C), and *Hygrocybe* (site D) were the fungal genera with the highest relative abundance in 20 cm at high altitude (2,119 m) (Fig. 5; Fig. S6).

**Co-occurrence analysis of the fungal community.** We performed network analysis to explore the co-occurrence patterns in the soil fungal community. The co-occurrence network consisted of 66 nodes and 113 edges, with 41.59% negative and 58.41% positive connections (Fig. 6A). Ascomycota was dominant in the network, with the largest numbers (37.88%), whereas Sebacinaceae (OTU38) and Boletaceae (OTU63) harbored a higher node degree, stress, and betweenness (data not shown), indicating that the two

**TABLE 1** Relationships of fungal richness and diversity with soil properties based on Spearman rank correlation analysis

| Soil property | Altitude gradient fungal richness or diversity[a] | | | | | | | | | | | |
| | 980 m | | 1,073 m | | 1,512 m | | 1,658 m | | 2,119 m | | Total | |
| | Richness | Diversity | Richness | Diversity | Richness | Diversity | Richness | Diversity | Richness | Diversity | Richness | Diversity |
|---|---|---|---|---|---|---|---|---|---|---|---|---|
| pH | −0.2945 | 0.0065 | 0.1022 | −0.1777 | 0.2350 | −0.0257 | −0.2095 | −0.0667 | −0.0975 | 0.0184 | −0.1417 | −0.0428 |
| OM | −0.2031 | 0.0159 | 0.0724 | −0.1908 | −0.0935 | −0.2074 | −0.0058 | 0.0917 | −0.1575 | −0.2181 | −0.1902* | −0.1507* |
| WC | −0.4013* | −0.0589 | 0.0504 | −0.0045 | 0.2974 | −0.0807 | 0.0714 | −0.0466 | −0.5775*** | −0.2792 | −0.3371*** | −0.1665* |
| TN | −0.3637 | −0.2480 | 0.0492 | −0.2254 | −0.1243 | −0.2776 | −0.0241 | 0.0388 | −0.1588 | −0.2358 | −0.2716*** | −0.2010** |
| TP | −0.1879 | −0.2933 | 0.1201 | 0.0118 | −0.1846 | −0.3134 | −0.0436 | −0.0868 | −0.1410 | 0.0347 | 0.2183** | 0.0861 |
| TK | 0.0224 | −0.0575 | −0.0093 | 0.2601 | 0.1677 | 0.0478 | −0.0179 | −0.1173 | 0.1328 | 0.1506 | −0.1339 | −0.0450 |
| AN | −0.1544 | 0.1382 | −0.0101 | −0.1592 | −0.0828 | −0.1521 | −0.0218 | 0.0911 | −0.1291 | −0.2006 | −0.2142** | −0.1151 |
| AP | −0.3019 | −0.0319 | 0.1337 | −0.1640 | −0.1526 | −0.2874 | −0.0475 | 0.0319 | −0.1440 | −0.1442 | 0.0707 | −0.0307 |
| AK | −0.1990 | 0.0920 | 0.1008 | −0.1243 | −0.1124 | −0.0886 | −0.0270 | 0.0823 | −0.0785 | −0.2029 | −0.1394 | −0.1259 |

[a]Significant correlation: *, $P < 0.05$, **, $P < 0.01$; ***, $P < 0.001$.

fungi might play a central role in the co-occurrence network. Meanwhile, we demonstrated the most densely connected nodes from the co-occurrence network based on the MCODE (i.e., Molecular Complex Detection) analysis and found that *Sebacina* and Boletaceae (optimal taxonomic annotations) represented a strong positive-interaction relationship (Fig. 6B).

**Evaluation of relationships and contributions of soil properties to fungal communities.** Soil properties had distinctive relationships with soil fungal richness and diversity (Table 1). We found that five soil properties (OM, WC, TN, TP, and AN) dramatically altered the fungal richness and diversity along the altitude gradient. Soil OM ($R^2 = −0.1902$ and $R^2 = −0.1507$), WC ($R^2 = −0.3371$ and $R^2 = −0.1665$), TN ($R^2 = −0.2716$ and $R^2 = −0.2010$), and AN ($R^2 = −0.2142$ and $R^2 = −0.1151$) showed a highly negative correlation with fungal richness and diversity, whereas TP ($R^2 = 0.2183$) showed a positive correlation with fungal richness. In addition, WC showed negative correlations with fungal richness at 980 m ($R^2 = −0.4013$) and 2,119 m ($R^2 = −0.5775$), indicating that WC might be the crucial indicator for the soil fungal characteristics of *T. sutchuenensis*. To further determine which fungal communities were affected by the five soil indicators identified (i.e., OM, WC, TN, TP, and AN), we then used an RF model to evaluate the contributions of soil properties to different fungal species. The RF results showed that soil OM was the primary variable for Zygomycota; soil WC was the dominant variable for Zygomycota, Ascomycota, and Basidiomycota. When considering the soil nutrients, soil TN and AN were the variables for Ascomycota, Zygomycota, and Basidiomycota; soil TP was the variable for Rozellomycota (Fig. 7).

## DISCUSSION

*T. sutchuenensis* is one of the endangered species unique to southwest China (i.e., Daba Mountains) and primarily occurs in the crevices of cliffs and rocks. Due to the hardwood, high oil content, and special odor, *T. sutchuenensis* has become a valuable plant in the country (6, 34). However, owing to human activities such as land development and deforestation, *T. sutchuenensis* has once again become an endangered species. There is a tight contact surface between plant roots and soil, and frequent material exchanges (e.g., water, heat, oxygen, and nutrients) strongly influence each other (10–12). Therefore, the soil is an important ecological factor for plants, and the growth of plants can be affected by controlling soil factors. To our knowledge, there have not been previous efforts to investigate soil properties and soil fungal communities at different altitudes (800 to 2,200 m) in southwest China, particularly at local scales. Taken together, the objective of this article is to present valuable and ground-based data to identify soil characteristics of the *T. sutchuenensis* across all known extant geographical ranges and to articulate multispatial distribution patterns of fungal communities in southwest China's mountainous areas.

Our results showed that the soil properties (i.e., pH, OM, WC, TN, TP, TK, AN, AP, and AK) displayed significant differences in soils at different altitudes and vertical depths.

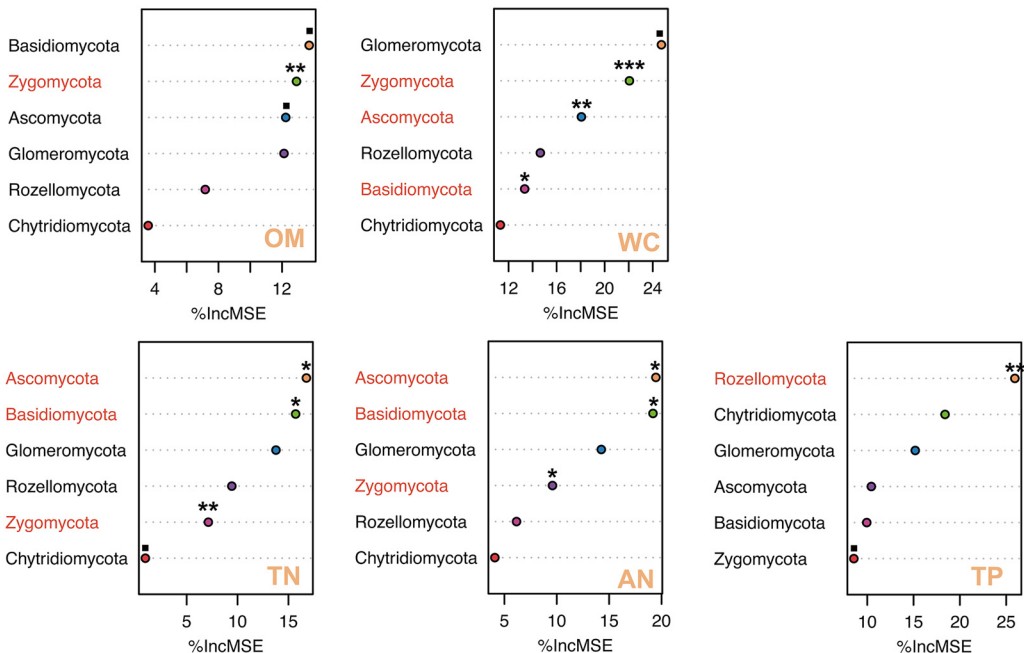

**FIG 7** Random forest (RF) analysis showing putative drivers of variation in soil properties. The percentage of increase of mean square error (IncMSE) of primary phyla was used to estimate the potential driver of soil properties. The accuracy was counted for individual trees and averaged over the whole forest (5,000 trees). Significance: ■, $P < 0.1$; *, $P < 0.05$; **, $P < 0.01$; ***, $P < 0.001$.

Still, there were no alterations at various horizontal distances. Soil is a dynamic system with multiple materials, and soil fertility is the foundation of the land's productivity (35, 36). Soil OMs, such as animal and plant residues, products of different decomposition stages of microorganisms, and humus produced by decomposition, are the primary indicators of soil fertility (14). Importantly, available N, P, and K are critical nutrients required for plant growth, which can be quickly absorbed by plants (37, 38). In this study, the soil fertility indicators, including OM, WC, TN, AN, TK, and AK, were higher in the two high-altitude soil samples (2,119 m and 1,658 m) than those at low altitudes (i.e., 1,073 m and 980 m). On the contrary, the contents of TP and AP decreased at 2,119 m and 1,658 m. Altitude gradients could induce changes in many environmental factors such as OM, WC, and nutrients (e.g., N, P, and K). An investigation of soil characteristics in mountainous areas in the southwestern United States found that OM, total soil porosity, and soil water-holding capacity increase with altitude (39). Also, the increased AN and AK contents in high-altitude areas are assumed to help *T. sutchuenensis* grow in high-altitude soils, providing the requisite nutrients (40). N associated with OM is not readily mineralized due to low temperatures; thus, relatively high TN of the soil at high altitude could result from high OM, which is similar to previous findings (41). A previous study found that TP was lower in Gangotri (3,415 m) than Kandakhal (1,532 m) due to the low-temperature-induced decrease in P mineralization (42), which was similar to our findings that high-altitude soils (2,119 m and 1,658 m) harbored lower TP and AP contents. In addition, it is well documented that AMF, such as Glomeromycota, are essential in the P cycles of the ground (43). Interestingly, Glomeromycota showed higher abundance at 980-m altitudes and the lowest abundance at 2,119-m altitudes (see Fig. S4 in the supplemental material), indicating that Glomeromycota might mediate the P cycles under different altitudinal gradients.

To further explore the effects of spatial variations on the soil fungal diversity, we then conducted alpha and beta analyses of the yielded data sets. Our results illustrated that altitude reduced the diversity of soil fungal communities and shaped the structure of soil fungal communities. Previous studies showed that the abundance and phylogenetic diversity of microbial groups in the Rocky Mountains of Colorado (at an altitude of 2,400 to 3,600 m) decrease with the increase in altitude (39). Similarly, at alpine

(2,300 to 2,530 m) and subalpine (1,500 to 1,900 m) altitudes, soil microbial activity also decreased with increasing altitude (44). In terms of vertical depth, the community diversity between soil samples at different depths changed rapidly because topsoil's and deep soil's physical and chemical properties are dramatically distinct (45). Nonetheless, the present study identified that vertical depth did not affect soil fungal community diversity and composition, which was partially explained by the fact that spatial variables on the fine scale (<50 cm) were much lower than the altitudinal gradient (46). Also, the *T. sutchuenensis* population thrives on cliffs, where there is little competition from other species, thus resulting in low genetic diversity and highly conservative microbial composition around the tree (47).

Functional fungi, such as EMF, AMF, and PPF, are abundant in soils. Among them, EMF amount to hundreds of species, including *Boletus*, *Clavulina*, *Cortinarius*, *Russula*, *Sebacinaceae*, and *Tomentella* (48). The trees with ectomycorrhizal roots include *Quercus*, *Pinus*, *Tilia*, *Juglans*, and *Betula* (49). For example, *Boletus edulis* and *Tricholoma matsutake* are often found in pine forests or spruce forests (50). Growing evidence has proved that EMF are beneficial to the growth of trees in poor and barren soil, promoting nutrient cycling and inducing systemic resistance against insects on a nonmycorrhizal plant (51). In this study, we found a great number of EMF in the soil samples, with an average of 26.89% (OTU richness). In the process of sampling, we noticed that many pine, oak, tilia, and walnut trees live at high altitude, whereas *Thuja*, *Quercus*, and *Cyclobalanopsis* primarily dominate at low altitude (Table S1), which was similar to the previous study of the forest community structure of *T. sutchuenensis* (8). Therefore, it is reasonable to assume that EMF are more abundant in high-altitude areas because of the existence of EMF-hosted species (i.g., pine, oak, tilia, and walnut), which could create beneficial conditions for the survival of *T. sutchuenensis* at high altitudes.

To further identify the dominant fungal species at a fine scale, we investigated the changes of fungal community response at the genus level in multispatial dimensions (Fig. 4; Fig. S10). In the topsoil of the rhizosphere (A) (20 cm), the dominant fungal genus *Tomentella* changed to *Piloderma* with increased altitudes (*Tomentella*→*Hygrocybe*→*Russula*→*Cuphophyllus*→*Piloderma*). Similarly, for the deep soil (60 cm) in the rhizosphere, the main dominant genus changes from *Tomentella* to *Cortinarius* (*Tomentella*→*Hygrocybe*→*Ramariopsis*→*Clavulinopsis*→*Cortinarius*). *Tomentella* is an ectomycorrhizal companion of many plants, including willows and alders, with abilities to help plants to overcome adverse stress (52, 53). The soil properties (except pH and P content) of *T. sutchuenensis* at middle and low altitudes were lower than those at high altitudes, so *Tomentella* might exist because of the high abundance of *Quercus* and *Cyclobalanopsis* that could help improve soil properties and promote *T. sutchuenensis* to cope with an adverse environment. *Hygrocybe*, frequently considered to be saprotrophic, can be found in a variety of habitats worldwide. While the exact trophic lifestyle of this genus remains unclear, it is possible that, considering the preponderance of *Hygrocybe* fungi in 1,073-m soils and their putative ability to act as root and systemic endophytes, these fungi may create ecologically essential symbiotic relationships with *T. sutchuenensis*, similar to an iconic conifer giant sequoia (*Sequoiadendron giganteum*) (54, 55). The fungus *Piloderma* can grow on rock surfaces, accelerating rock decomposition and weathering, obtaining essential nutrients and releasing minerals (56, 57). The high abundance of *Piloderma* indicated that it might contribute to the initial establishment of *T. sutchuenensis* in mountainous areas, and the differentiation pattern of topsoil fungal communities with altitude might be a strategy for habitat adaptation of *T. sutchuenensis*. Some members of *Cortinarius* play a key role in mobilizing nutrients from organic matter and accelerating N cycling at high altitudes (58, 59). With elevation, the soil physicochemical properties (i.e., OM, WC, N, and K) increased, so that the fungal community changed from stress-resistant taxa (i.e., *Tomentella*) to vegetative taxa (i.e., *Piloderma* and *Cortinarius*).

*Sebacina* (Sebacinaceae) and Boletaceae were the two highly connected co-occurrence fungal taxa, and they presented a strong positive interaction. *Sebacina* (Sebacinaceae) is well documented as an efficient fungus for promoting tree productivity and stress resistance (60, 61). Plants colonized by Sebacinaceae have increased mineral nutrient

acquisition, resulting in growing elevated N, P, and K contents (62, 63). The Boletaceae are a group of wild and edible mushroom fungi existing in most forest ecosystems, which can generate mycorrhiza with the roots of oak and pine and enhance the absorption of water and mineral nutrients by Masson pine (64). Based on the field investigation, we found that many trees with ectomycorrhizal roots, such as *Quercus*, *Pinus*, *Tilia*, *Juglans*, and *Betula*, existed in the sampling sites. Previous studies showed that Boletaceae have distinctive connections with *Pinus* and *Quercus* (50, 65); thus, the dominance of Boletaceae might be induced by the host plants and help improve the cycle of nutrients. Although no previous studies showed that Sebacinaceae and Boletaceae have a strong correlation, as two types of beneficial fungi, it is reasonable to suggest that they could promote the growth and survival of *T. sutchuenensis* to a certain extent and, at the same time, improve the stress tolerance of *T. sutchuenensis*. These two co-occurrence fungi can be identified at all sampling sites, representing a strong conserved relationship and can be putative markers for the conservation of *T. sutchuenensis*.

To explore the relationship between soil fungi and soil physical and chemical properties, we used correlation analysis and random forest (RF) analysis of the fungal characteristics and dominant phyla, respectively. The variation of fungal richness and diversity can be largely explained by the close relationships with soil OM, WC, TN, TP, and AN. Particularly, WC showed significant negative correlations with fungal richness at both 980 m and 2,119 m. The soil WC is reported to be a crucial driver and indicator for fungal diversity and composition, especially in the semiarid mountain environment in China (66). Hence, soil WC might be an essential indicator for habitat investigation of *T. sutchuenensis* and root-related fungal characteristics. Interestingly, four soil variables (OM, WC, TN, and AN) could explain Zygomycota (Ascomycota and Basidiomycota can be partially explained), whereas TP was the main variable of Rozellomycota. *T. sutchuenensis* primarily lives in a natural environment; thus, plant residues and fallen leaves accumulated in the soil could convert into humus, increase the content of organic matter and nutrients, which is highly suitable for the growth of Zygomycota, which can make good use of the saprophytic environment (67). This finding is in agreement with some studies that have reported that Zygomycota species, such as *Absidia*, *Lichtheimia*, *Pilobolus*, *Rhizopus*, and *Syncephalastrum*, are distributed in semiarid areas and directly participate in the decomposition of fecal waste, involving in carbon, nitrogen, and energy cycles (68). Also, RF analysis identified that P content could explain the variation of Rozellomycota, similar to previous findings, which pointed out that Rozellomycota has a positive correlation with TP content (69). Notably, TP was the only factor that harbored a positive correlation with fungal richness. Together with our findings on fungal community composition responses, our results show that the fungal community's composition was primarily driven by alterations in WC and TP along the altitudinal gradients. This observation is in agreement with previous soil fungus and AMF studies that reported soil properties like TP and WC were essential markers along the elevation gradient (70).

It is worth noting that the current large-scale soil research is mainly focused on the microbial community composition and functional groups during the succession of boreal forests (71). Still, no analysis has been carried out in the mountainous areas of southwest China or the Daba Mountains. In this study, soil samples were taken at elevations between 800 and 2,200 m to fully explore the soil properties, fungal community composition and species diversity of mountainous ecosystems in southwest China. Hence, it is the first time that large-scale soil property and fungal community investigations have been carried out in southwest China, providing a basis for exploring the distribution pattern of regional soil microorganisms. Nevertheless, further studies should investigate the bacterial communities, as they are also crucial in participating in soil nutrient cycles and plant growth and survival.

**Conclusion.** We found that fungal diversity and community composition of soil samples were affected primarily by altitudinal gradient rather than the horizontal distances from a tree base and the vertical soil layers therein. On the same line, soil properties changed with the alteration of altitude gradient. The abundance of fungal genera at different altitudes was distinct: *Tomentella* and *Piloderma* were the dominant fungal genera at

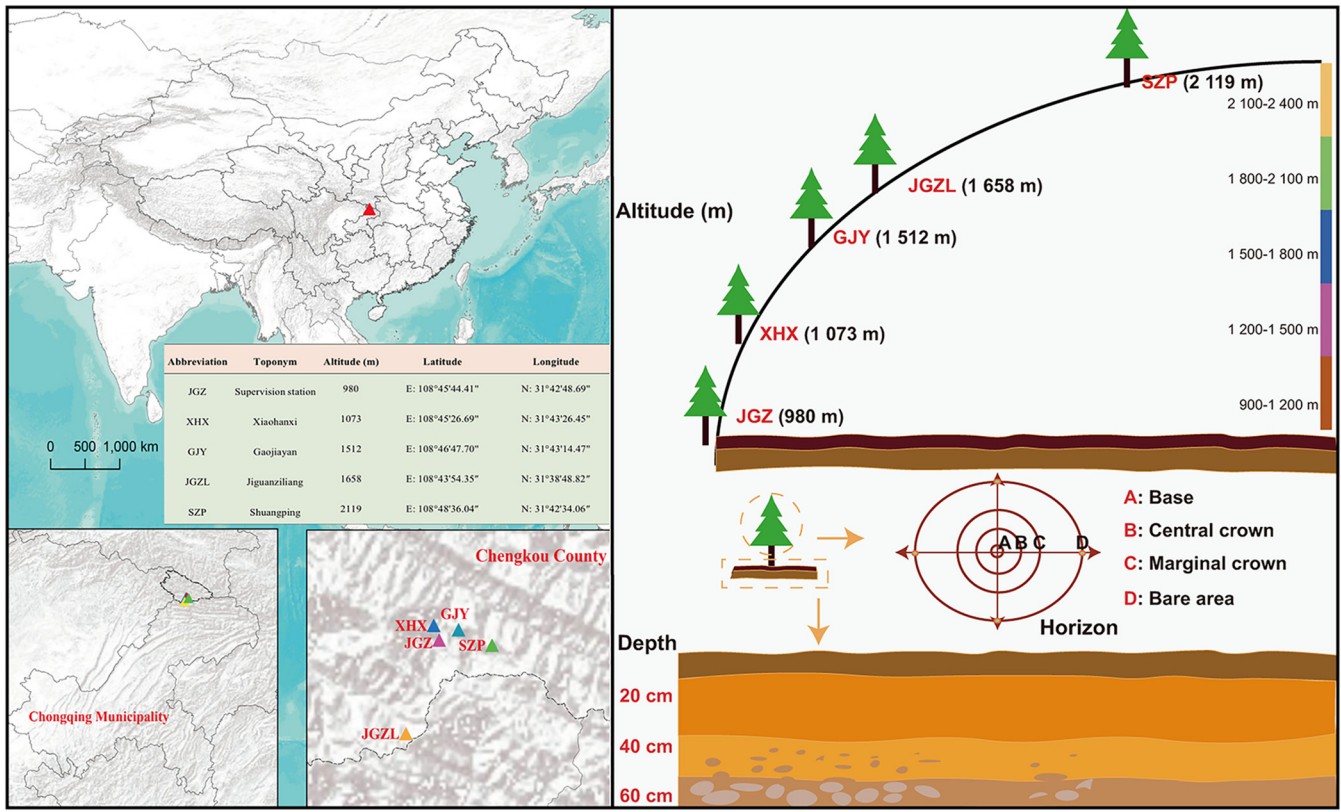

**FIG 8** Map of the study area and the sampling sites along the altitudinal gradient, horizontal distance, and vertical depth. The table in the map represents specific information about the sampling sites.

the lowest and highest altitudes in the topsoil, respectively, suggesting an enhancement in the stress resistance and rooting ability of *T. sutchuenensis*. The highest abundance of EMF and the lowest abundance of PPF could partially explain the spread of *T. sutchuenensis* to high elevations. Correlation analysis and RF analysis identified that soil WC and TP were the essential indicators of and contributors to the composition of soil fungi. In sum, these results highlight the importance of fungal communities of *T. sutchuenensis* and generate insights into the altitudinal pattern and drivers of fungal community in soil ecosystems.

## MATERIALS AND METHODS

**Community survey of *T. sutchuenensis*.** The present study used line transect methods and field investigation to investigate all known field communities of *T. sutchuenensis* (72). A previous study demonstrated that the tree's distribution is between the elevation interval of 800 and 2,100 m in the subtropical Daba Mountains (8). In this study, sampling sites were chosen every 300 m over an elevation gradient, for a total of five elevation gradients: 700 to 1,000 m, 1,000 to 1,300 m, 1,300 to 1,600 m, 1,600 to 1,900 m, and 1,900 to 2,200 m. Specifically, the sampling sites included the supervision station (108.7623°E, 31.7135°N, 980 m), Xiaohanxi (108.7574°E, 31.7240°N, 1,073 m), Gaojiayan (108.7799°E, 31.7209°N, 1,512 m), Jiguanziliang (108.7318°E, 31.6469°N, 1,658 m), and Shuangping (108.8100°E, 31.7095°N, 2,119 m) (Fig. 8). According to the elevation gradient, the representative altitudes were 980 m, 1,073 m, 1,512 m, 1,658 m, and 2,119 m. Meanwhile, according to the principle of similar topographic conditions, such as soil type and hill slope, three *T. sutchuenensis* trees (with similar sizes and shapes) were selected for each gradient (tree sampling plots).

The dominant plant species at five elevational gradients were recorded based on field investigation. For plant investigation, four 10-by 10-m quadrants, five 5- by 5-m quadrants, and two 1- by 1-m quadrants were randomly selected in each 20- by 20-m plot to determine the composition and richness of the trees, shrubs, and herbs, respectively. Interestingly, *Thuja*, *Quercus*, and *Cyclobalanopsis* were the dominant trees at 980 m, 1,073 m, and 1,512 m, *Pinus*, *Juglans*, and *Carpinus* were the main trees at 1,658 m, and *Quercus*, *Elaeagnus*, and *Pinus* were the primary trees at 2,119 m (see Table S1 in the supplemental material). Except for the 1,073-m group, which is near a river (Xiaohan River), the four species' diversity (Shannon, Simpson, Gleason, and Pielou) indexes were decreased with elevation.

**Soil sample collection.** As mentioned above, we identified the tree sampling plot from five elevation gradients. Within each tree sampling plot, to investigate the soil fungal community at a fine scale,

four sites were selected as horizontal soil collection points according to the distance between sampling points and the base of the trunk: near the base (A), the center of the crown (B), the edge of the crown (C), and the bare area outside the crown (D) (Fig. 8). Furthermore, vertical samples with depths of 0 to 20 cm, 20 to 40 cm, and 40 to 60 cm (labeled "20 cm," "40 cm," and "60 cm," respectively, on the figure) were taken for each horizontal site (Fig. 8). A total of 180 samples were collected from the five elevation gradients (three tree sampling plots, four horizontal distances, and three vertical depths) (Fig. 8). The soil samples were immediately placed in an insulated container with ice and then transported to the laboratory. After removing debris and roots, the soil samples were well ground and sieved (<2 mm). A portion of the soil samples was stored at −80°C for subsequent molecular analysis, whereas the remaining soil was used for soil physical and chemical characterization.

**Soil physical and chemical properties.** Soil physical and chemical analyses for pH, OM, WC, total nitrogen (TN), total phosphorus (TP), total potassium (TK), available nitrogen (AN), available phosphorus (AP), and available potassium (AK) were performed as previously reported, and details are given in the supplemental material.

**DNA extraction and high-throughput sequencing.** Microbial DNA from each soil sample (0.5 g) was extracted following the QIAmp DNA stool minikit manufacturer's instructions. To investigate the soil fungal community, the internal transcribed spacer 1 (ITS1) and ITS2 sequences were amplified with a barcoded ITS1 primer (73). PCR amplification was conducted with the following modifications: 94°C for 1 min, followed by 35 cycles of 94°C for 30 s, 52°C for 30 s, 68°C for 30 s, and 6°C for 10 min. Subsequently, high-quality samples were detected by 2% agarose gel electrophoresis and used as the templates for PCR amplification for 8 cycles. PCR products with a bright band were purified with GeneJET gel extraction kit according to the manufacturer's instructions (Thermo Scientific). An equal concentration of PCR product from each sample was sequenced on the Illumina MiSeq platform with 300-bp paired-end reads (Illumina, Inc., San Diego, CA, USA) at the TinyGene Bio-Tech Co., Ltd. (Shanghai, China).

**Sequence processing.** The reads obtained by Miseq sequencing were distinguished from each sample according to the barcode. Any low-quality sequences were removed by ultrafast sequence analysis (USEARCH) based on the UCHIME algorithm. The splicing sequence was qualified and filtered to yield the optimized sequence. The optimized sequence parameters were set to maxAMBIG = 0, maxHOMOP = 8, minLength = 200, maxLength = 580 (Mothur V.1.39.5). The optimized sequences were subjected to cluster analysis of operational taxonomic units (OTUs) at a 97% similarity level using the UPARSE pipeline (74). A total of 9,930,771 raw fungal reads were yielded, and a total of 8,933,216 clean reads were collected after quality filtering. OTU taxonomic classification was performed by searching the representative sequence set against UNITE+INSD version 5.0 (i.e., the UNITE and the International Nucleotide Sequence Databases). FUNGuild analysis was used to sort fungal sequence pools into three ecologically meaningful categories (i.e., pathotrophs, saprotrophs, and symbiotrophs) (46).

**Statistical analyses.** All statistical analyses were performed in the R environment (http://www.r-project .org/). Analysis of similarity (ANOSIM) was performed to analyze the similarities among multispatial data sets (i.e., altitudinal gradient, horizontal distance, and vertical depth). Chao1 and Shannon's indexes were utilized to estimate the alpha diversity of the three types of soil samples based on the sequence reads from the richness and diversity of community levels using Mothur software. Alpha diversity was compared via linear mixed-effects (LME) models by the "nlme" package. In order to explore the relationships between fungal communities and spatial variations (i.e., altitudinal gradient, horizontal distance, and vertical depth), beta diversity was calculated using two axes of a nonmetric multidimensional scaling (NMDS) analysis of Bray-Curtis dissimilarities in the OTUs' community matrix by using the "vegan" package.

Canonical discriminant analysis (CDA) was performed using the "candisc" package to identify the significant taxonomic alterations. To investigate the fungal taxa responsible for the community differentiation among the different altitudinal gradients, horizontal distances, and vertical depths, we conducted linear statistical analysis on all identified OTUs with the "limma" package. The indicator OTUs from the three types of soil samples were presented by ternary plots with the "ggtern" and "dplyr" packages (with only the most abundant four genera displayed).

To explore the co-occurrence patterns of fungal communities among all soil samples, we calculated rank correlation coefficients of taxonomic genera based on the Molecular Ecological Network Analysis Pipeline through random matrix theory (RMT)-based methods (MENA [http://ieg4.rccc.ou.edu/mena/]) (RMT threshold of 0.85, $P < 0.05$) (75). The visualization of the co-occurrence network was constructed by Cytoscape software (https://cytoscape.org/). To further identify closely interacted fungal genera among a large network, we applied Molecular Complex Detection (MCODE) analysis with a $k$ score of >5 to the yielded network.

Analysis of variance (ANOVA) was performed to analyze the significance of different comparisons of soil physical and chemical properties (i.e., pH, OM, WC, TN, TP, TK, AN, AP, and AK). The relationships between the soil fungal characteristics and the soil properties were determined using Spearman correlation analysis. In addition, random forest (RF) analysis was employed to identify the crucial driver of taxonomic phylum (>5% of total community) for the soil's physical and chemical properties with the "randomForest" package. Percentages of the increased mean squared error (IncMSE) of variables were utilized to analyze the contribution of the predictors. The relative abundance of OTUs was calculated by dividing the number of sequences contained in the OTUs by the total number of sequences. Significance levels are indicated on the figures as follows: ■, $P < 0.1$; *, $P < 0.05$; **, $P < 0.01$; ***, $P < 0.001$. Adobe Illustrator CC and Adobe Photoshop CS6 were used for figure processing.

**Data availability.** The data sets presented in this study can be found in online repositories. The names of the repository/repositories and accession numbers can be found at https://www.ncbi.nlm.nih

.gov/GenBank. The soil fungal data set was deposited in the NCBI Sequence Read Archive under accession no. PRJNA774433.

## SUPPLEMENTAL MATERIAL

Supplemental material is available online only.

**SUPPLEMENTAL FILE 1**, PDF file, 1.3 MB.

## ACKNOWLEDGMENTS

We thank the anonymous reviewers for their careful work and thoughtful suggestions, which helped improve this paper substantially. We thank Bo Lv (Hunan Normal University) and Jian-ping Xie (Southwest University) for technical support, as well as the staff of Chongqing Dabashan National Nature Reserve Management Affairs Center and local farmers for their assistance in field sampling.

This study was funded by the Chongqing Technology Innovation and Application Development Special Key Project—Research and Application of Typical Damaged Ecosystem Restoration Technology in Nature Reserve (cstc2019jscx-tjsbX0005) and 2021 National Key National Protection of Wild Animals and Plants Project of the Central Forestry Reform and Development Fund - Research on the Protection of National Key Wild and Endangered Plants in the Chongqing Section of the Upper Yangtze River (20211210).

Y.-w. Zuo and H.-p. Deng conceived and designed the study. P. He, J.-h. Zhang, C.-y. Xia, and H. Zhang helped with the experiment design. Y.-w. Zuo, D.-h. Ning, Y.-l. Zeng, W.-q. Li, and Y. Yang collected the samples. Y.-w. Zuo and D.-h. Ning analyzed the data and prepared the figures and table. H.-p. Deng and J.-h. Zhang helped with the improvement of the manuscript. All authors read and approved the final manuscript.

We declare no conflict of interest.

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
