## [Reviewer comments · Microbiology Spectrum]

Microbiology Spectrum

Contrasting responses of multi-spatial soil fungal communities of *Thuja sutchuenensis* Franch., an extremely endangered conifer in southwestern China

You-wei Zuo, Jia-hui Zhang, Deng-hao Ning, Yu-lian Zeng, Ying Yang, Chang-ying Xia, Huan Zhang, and Hong-ping deng

Corresponding Author(s): Hong-ping deng, Southwest University

Review Timeline:

Submission Date:	January 21, 2022
Editorial Decision:	April 24, 2022
Revision Received:	May 8, 2022
Accepted:	May 20, 2022

Editor: Chengshu Wang

Reviewer(s): The reviewers have opted to remain anonymous.

Transaction Report:

DOI: <https://doi.org/10.1128/spectrum.00260-22>

April 24, 2022

Prof. Hong-ping deng
Southwest University
No.2 Tiansheng Road Beibei District, Chongqing 400715, P.R. China
chongqing
China

Re: Spectrum00260-22 (**Contrasting responses of multi-spatial soil fungal communities of *Thuja sutchuenensis* Franch., an extremely endangered conifer in southwestern China**)

Dear Prof. Hong-ping deng:

Link Not Available

Sincerely,

Chengshu Wang

Journals Department
Reviewer comments:

Reviewer #1 (Comments for the Author):

Zuo et al Functional Ecology review 2022

This study focuses on characterizing the soil fungal communities present in areas where the endangered conifer *Thuja sutchuenensis* naturally occurs in southwestern China. This imperiled conifer is one of high conservation value and knowledge about *Thuja*-associated microbial communities relative to many other trees is low, hence knowledge about the soil fungal communities associated with *T. sutchuenensis* does have notable value. The study uses a molecular-based approach in which soil cores are collected in the vicinity of *Thuja* stems at five different elevations. The molecular techniques and bioinformatic

analyses are standard for these kinds of microbiome studies. In general, the writing is clear and free of technical errors, although many sections could be condensed for greater readability. The figures are informative and the supplementary material is well organized.

The results in terms of both soil nutrient conditions and general soil alpha and beta diversity patterns are consistent with many previous studies, in which nutrient availability and community richness and composition change with elevation.

While I think there is value in this work, the current conclusions have to me a fatal flaw. The results clearly suggest the most responsive and important fungal taxa in these soil communities are ectomycorrhizal (ECM) in lifestyle. While I am not surprised about that since ECM fungi often have high abundance in forest soils, Thuja is not an ECM host in any previous study I am aware of. I checked the FungalRoot global database (Soudziloskaia et al. 2020), it is very clear that Thuja is only a host of arbuscular mycorrhizal (AM) fungi. Given this - there is a real problem with emphasizing that much of the fungal community associated with Thuja is not coming from Thuja.

The only path forward I see to make these results publishable is to both characterize the full vegetation community present at each sampling location (see below) and also to microscopically examine the roots of Thuja sutchunensis and determine mycorrhizal status. I am 100% confident the microscopic analysis will confirm it is an AM host and what is likely driving the trends in EM fungi is differences in EM host tree composition with elevation.

Once that is done, I think a full-scale revision to the writing must be made in which it is very clearly indicated that Thuja is not the host of any of ECM taxa currently highlighted in the abstract and main text. This is not only important for scientific accuracy in the results, but also if the authors are at all serious about trying to have this data used for conservation purposes. For example, if Thuja is really dependent on these ECM taxa to thrive, then reforestation efforts will require co-planting of both Thuja and EM host trees.

L167 - While the specific details about Thuja and how it is sampled are sufficient here, there is no information about any of the additional plant community at any of these sites. This is a huge problem, as it is unclear the extent to which the soil fungal communities profiled are really reflective of Thuja or are influenced by additional plant hosts. At a bare minimum, there should be clear information about co-occurring tree species at each site and elevation along with details about the herbaceous layer. A much better presentation would be to specifically quantify the vegetation in the immediate 10-20 m surrounding each sampling tree at each site to determine the plant neighborhood present (with say the basal area of each tree species clearly presented).

L177 - There are five elevations in this study, so I think the total number of samples was $180 \times 5 = 900$. Did I miss understand?

L179 - Was this soil sieved? It is unclear to the extent these extractions include just soil or also include roots. If roots are included, this could heavily skew the sequence read counts.

L192 - It is not possible to amplify both the ITS1 and ITS2 with a single barcoded primer set using MiSeq technology. My guess is only the ITS1 region was amplified - this should be clarified.

Having identified the fatal flaw of this study early on, I only quickly perused the discussion. I find many parts to be misguided, particularly in linking ECM taxa to Thuja performance.

Staff Comments:

Preparing Revision Guidelines

For complete guidelines on revision requirements, please see the journal Submission and Review Process requirements at <https://journals.asm.org/journal/Spectrum/submission-review-process>. **Submissions of a paper that does not conform to**

Microbiology Spectrum guidelines will delay acceptance of your manuscript. "

Please return the manuscript within 60 days; if you cannot complete the modification within this time period, please contact me. If you do not wish to modify the manuscript and prefer to submit it to another journal, please notify me of your decision immediately so that the manuscript may be formally withdrawn from consideration by Microbiology Spectrum.

1 Dear Professor (s),

2 Thank you very much for your letter dated on Apr 25, 2022, and the referees'
3 reports. Based on your comment and request, we have made relevant modifications on
4 the manuscript. Here, we attached a revised manuscript for your approval. A
5 document answering every question from the referees was also summarized and
6 enclosed.

7 A revised manuscript with the correction sections color marked was attached as
8 the supplemental material for the purpose of easy checking and editing. Some
9 grammatical or typographical errors have been corrected. All the lines and pages
10 indicated above are in the revised manuscript.

11 Should you have any questions, please contact us without hesitate.

12 Thank you and all the reviewers for the kind advice.

13 Sincerely yours,

14 Hong-ping Deng

15

16 **Editor and Reviewer comments:**

17 *Reviewer #1 (Comments for the Author):*

18 *Zuo et al Functional Ecology review 2022*

19 *This study focuses on characterizing the soil fungal communities present in areas*
20 *where the endangered conifer Thuja sutchuenensis naturally occurs in southwestern*
21 *China. This imperiled conifer is one of high conservation value and knowledge about*
22 *Thuja-associated microbial communities relative to many other trees is low, hence*
23 *knowledge about the soil fungal communities associated with T. sutchuenensis does*
24 *have notable value. The study uses a molecular-based approach in which soil cores*
25 *are collected in the vicinity of Thuja stems at five different elevations. The molecular*
26 *techniques and bioinformatic analyses are standard for these kinds of microbiome*
27 *studies. In general, the writing is clear and free of technical errors, although many*
28 *sections could be condensed for greater readability. The figures are informative and*
29 *the supplementary material is well organized.*

30 *The results in terms of both soil nutrient conditions and general soil alpha and*
31 *beta diversity patterns are consistent with many previous studies, in which nutrient*
32 *availability and community richness and composition change with elevation.*

33 *While I think there is value in this work, the current conclusions have to me a*
34 *fatal flaw. The results clearly suggest the most responsive and important fungal taxa*
35 *in these soil communities are ectomycorrhizal (ECM) in lifestyle. While I am not*
36 *surprised about that since ECM fungi often have high abundance in forest soils, Thuja*
37 *is not an ECM host in any previous study I am aware of. I checked the FungalRoot*
38 *global database (Soudziloskaia et al. 2020), it is very clear that Thuja is only a host*
39 *of arbuscular mycorrhizal (AM) fungi. Given this - there is a real problem with*
40 *emphasizing that much of the fungal community associated with Thuja is not coming*
41 *from Thuja.*

42 *The only path forward I see to make these results publishable is to both*
43 *characterize the full vegetation community present at each sampling location (see*
44 *below) and also to microscopically examine the roots of Thuja sutchunensis and*
45 *determine mycorrhizal status. I am 100% confident the microscopic analysis will*
46 *confirm it is an AM host and what is likely driving the trends in EM fungi is*
47 *differences in EM host tree composition with elevation.*

48 *Once that is done, I think a full-scale revision to the writing must be made in*
49 *which it is very clearly indicated that Thuja is not the host of any of ECM taxa*
50 *currently highlighted in the abstract and main text. This is not only important for*
51 *scientific accuracy in the results, but also if the authors are at all serious about trying*
52 *to have this data used for conservation purposes. For example, if Thuja is really*
53 *dependent on these ECM taxa to thrive, then reforestation efforts will require*
54 *co-planting of both Thuja and EM host trees.*

55 **Answer1:** We deeply appreciate your careful work and thoughtful suggestions!
56 Also, we sincerely apologize for our honest mistake about the description of ECM
57 because we carelessly deleted the explanation of ECM in the last manuscript. Yes,
58 *Thuja* is not an ECM host in any previous study, but we found many trees with
59 ectomycorrhizal roots, such as *Quercus*, *Pinus*, *Tilia*, *Juglans*, and *Betula*, were
60 existed in the sampling sites based on the field investigation. Hence, we assumed that
61 the ECM taxa identified in the soil samples around *Thuja* might be induced by these
62 ECM-hosted plants. Accordingly, we have re-added the missed key points in the main
63 document (see methods and discussion) and supplementary file (see table s1). Again,
64 we thank you for your valuable comments and hope the augment will strengthen the
65 manuscript.

66 Changes: Line 161-170 “The dominant plant species...were decreased with
67 elevation.” was added.

68 Line 444-459 “Functional fungi, such as EMF, ... in high altitudes.” was added.

69 Table S1 was added in the supplementary file.

70

71 *L167 - While the specific details about Thuja and how it is sampled are sufficient*
72 *here, there is no information about any of the additional plant community at any of*
73 *these sites. This is a huge problem, as it is unclear the extent to which the soil fungal*
74 *communities profiled are really reflective of Thuja or are influenced by additional*
75 *plant hosts. At a bare minimum, there should be clear information about co-occurring*
76 *tree species at each site and elevation along with details about the herbaceous layer.*
77 *A much better presentation would be to specifically quantify the vegetation in the*
78 *immediate 10-20 m surrounding each sampling tree at each site to determine the plant*
79 *neighborhood present (with say the basal area of each tree species clearly presented).*

80 **Answer2:** Yes, this is a valid question and we do believe that the information
81 about plant community is crucial for better understanding the soil fungal communities.
82 As mentioned in Answer1, we apologize for the honest mistake and have added vital
83 information in the manuscript.

84 Changes: Line 161-170 “The dominant plant species...were decreased with

85 elevation.” was added.

86 Table S1 was added in the supplementary file.

87

88 *L177 - There are five elevations in this study, so I think the total number of*
89 *samples was $180 \times 5 = 900$. Did I miss understand?*

90 **Answer3:** Thank you for your kind comments! The samples were from five
91 elevational gradients, four horizontal distances, three vertical depths, and three
92 biological replicates ($5 \times 4 \times 3 \times 3 = 180$).

93

94 *L179 - Was this soil sieved? It is unclear to the extent these extractions include*
95 *just soil or also include roots. If roots are included, this could heavily skew the*
96 *sequence read counts.*

97 **Answer4:** We appreciate your professional suggestions and have added details
98 about the treatment of soil samples. Specifically, the collected soil samples were
99 immediately placed in an insulated container with ice and then transported to the
100 laboratory. After removing debris and roots, the soil samples were well ground and
101 sieved (<2 mm).

102 Changes: Line 180-183 “The soil samples...well ground and sieved (<2 mm).”
103 was added.

104

105 *L192 - It is not possible to amplify both the ITS1 and ITS2 with a single*
106 *barcoded primer set using MiSeq technology. My guess is only the ITS1 region was*
107 *amplified - this should be clarified.*

108 **Answer5:** Thank you for your helpful comments! We have corrected the
109 sentence accordingly.

110

111 *Having identified the fatal flaw of this study early on, I only quickly perused the*
112 *discussion. I find many parts to be misguided, particularly in linking ECM taxa to*
113 *Thuja performance.*

114 **Answer6:** We acknowledge your comments and have elaborated on the ECM
115 taxa more accurately in the revised manuscript. We deeply appreciate your
116 professional work that helped improve this paper substantially.

117 Line 470-472 “...so *Tomentella* might ...with an adverse environment. ” was
118 rephrased.

119 Line 497-501 “Based on the field investigation,...improve the cycle of nutrients”
120 was added.

121

122 Other changes:

123 Following suggestions, we have listed references in the order and presented
124 figures as source files (individual TIFF). Several other typos and linguistic errors have
125 been corrected.

126

127 **Acknowledgements**

128 We acknowledge the reviewer’s comments and suggestions very much, which are
129 valuable in improving the quality of our manuscript. Should you have any questions,
130 please contact us without hesitation.

131

132 Thank you for the kind advice!

133

134 Sincerely yours,

135 Hong-ping Deng

May 20, 2022

Prof. Hong-ping deng
Southwest University
No.2 Tiansheng Road Beibei District, Chongqing 400715, P.R. China
chongqing
China

Re: Spectrum00260-22R1 (**Contrasting responses of multi-spatial soil fungal communities of *Thuja sutchuenensis* Franch., an extremely endangered conifer in southwestern China**)

Dear Prof. Hong-ping deng:

Your manuscript has been accepted, and I am forwarding it to the ASM Journals Department for publication. You will be notified when your proofs are ready to be viewed.

Sincerely,

Chengshu Wang
Editor, Microbiology Spectrum
